# Investigating the Use of SARS-CoV-2 (COVID-19) Odor Expression as a Non-Invasive Diagnostic Tool—Pilot Study

**DOI:** 10.3390/diagnostics13040707

**Published:** 2023-02-13

**Authors:** Janet Crespo-Cajigas, Vidia A. Gokool, Andrea Ramírez Torres, Liam Forsythe, Benjamin S. Abella, Howard K. Holness, Alan T. Charlie Johnson, Richard Postrel, Kenneth G. Furton

**Affiliations:** 1Global Forensic and Justice Center, Department of Chemistry and Biochemistry, Florida International University, Miami, FL 33199, USA; 2Department of Emergency Medicine and Penn Acute Research Collaboration, University of Pennsylvania, Philadelphia, PA 19104, USA; 3Department of Physics and Astronomy, University of Pennsylvania, Philadelphia, PA 19104, USA; 4VOC Health, Inc., Miami Beach, FL 33140, USA

**Keywords:** SARS-CoV-2, COVID-19, odor signature, HS-SPME-GC-MS, machine learning, sPLS-DA modeling, non-invasive diagnostic tool

## Abstract

Since the beginning of the COVID-19 pandemic, there has been enormous interest in the development of measures that would allow for the swift detection of the disease. The rapid screening and preliminary diagnosis of SARS-CoV-2 infection allow for the instant identification of possibly infected individuals and the subsequent mitigation of the disease spread. Herein, the detection of SARS-CoV-2-infected individuals was explored using noninvasive sampling and low-preparatory-work analytical instrumentation. Hand odor samples were obtained from SARS-CoV-2-positive and -negative individuals. The volatile organic compounds (VOCs) were extracted from the collected hand odor samples using solid phase microextraction (SPME) and analyzed using gas chromatography coupled with mass spectrometry (GC-MS). Sparse partial least squares discriminant analysis (sPLS-DA) was used to develop predictive models using the suspected variant sample subsets. The developed sPLS-DA models performed moderately (75.8% (±0.4) accuracy, 81.8% sensitivity, 69.7% specificity) at distinguishing between SARS-CoV-2-positive and negative -individuals based on the VOC signatures alone. Potential markers for distinguishing between infection statuses were preliminarily acquired using this multivariate data analysis. This work highlights the potential of using odor signatures as a diagnostic tool and sets the groundwork for the optimization of other rapid screening sensors such as e-noses or detection canines.

## 1. Introduction

Severe acute respiratory syndrome coronavirus 2 (SARS-CoV-2) has resulted in the deaths of over 6 million people on a global scale since the start of the COVID-19 pandemic in 2019 [1]. Due to the widespread expansion of this disease, along with its high capacity for infection even among asymptomatic individuals, there have been multiple studies on the development of technology and tools that would aid in its rapid screening and diagnosis [2,3,4,5]. The most common tools currently used for confirmatory diagnosis are the reverse transcriptase–quantitative polymerase chain reaction (RT-qPCR) test, which has been known to have ~90% sensitivity, on average [6,7,8], and the antigen test, which varies widely in terms of sensitivity and can range from 35 to 72% depending on the brand [6,7,9]. Despite the high sensitivity RT-qPCR is known for, there are several disadvantaging factors that can lead to false negatives such as a degraded sample or inadequate sample collection, a limit of detection higher than the available viral content, viral mutations, and a poor performance in sample testing [10]. False negative tests can result in further spreading of the disease by undiagnosed patients. Additionally, PCR-based tests seem to have a long turnaround time, with individuals receiving a notification of their test results 24–48 h after specimen collection, further enhancing viral circulation. On the other hand, antigen tests are more efficient for patient screening, as they produce swift responses, are less expensive, and allow for self-testing [7,9]. Unfortunately, the sensitivity of the assay is sacrificed in exchange for a rapid response time. The sensitivity is even further reduced in asymptomatic populations [9]. Moreover, both RT-qPCR and antigen diagnostic tests can require a marginally invasive and uncomfortable sample collection from the patient [11], which could discourage individuals from seeking diagnostic services. There remains a pressing need for effective screening and diagnostic testing for public health surveillance, as SARS-CoV-2 has been proven to cause severe health outcomes.

Some other methods of detection that have been preliminarily conceptualized include—but are not limited to—the use of analytical instrumentation [4,12], e-noses [13], and canines [4,12,14]. Each of the aforementioned research studies have implemented the analysis of volatile organic compounds (VOCs) that are characteristic of SARS-CoV-2-infected individuals. VOCs are emitted by humans and other living organisms through various bodily secretions such as breath, saliva, urine, and sweat [15,16]. These VOCs can change over time and are often influenced by alterations in metabolic conditions, such as those induced by diseases [17]. Multiple reviews have been published over the past decade on the scent of diseases and how odor expression can potentially be used to diagnose various ailments such as gastro-intestinal diseases, infected wounds, and cancer [17,18,19]. Recently, the diagnosis of SARS-CoV-2 infection through the analysis of exhaled breath VOCs has become more thoroughly explored as an alternative screening method [3,20,21,22]. In 2020, Ruszkiewicz et al. performed a preliminary study wherein they discovered that they were able to distinguish between individuals who were SARS-CoV-2-infected and those who were presenting similar symptoms using gas chromatography (GC) and ion mobility spectroscopy with approximately 80.7% accuracy [22]. They hypothesized that a series of compounds including methanol and various ketones could provide a foundation for the development of a breath test for SARS-CoV-2 infection. Another study in 2021 commented on the reduced concentrations of acetone in the breath samples of SARS-CoV-2-infected individuals [20]. Additionally, Abumeeiz et al. (2021) assessed the opportunities and challenges of using a breathalyzer for the diagnosis of SARS-CoV-2 infections. The authors remarked that while high upfront costs and standardization might become an issue, the implementation of this technology could provide a worthwhile, noninvasive, and rapid test that would enable better disease management and responses to community outbreaks [3]. The researchers also took on a different approach to collect VOC odor from patients in a less invasive fashion, that is, from the hands versus the breath. Numerous studies have been published on the availability of VOCs from the hands of individuals and the ability of these VOCs to differentiate between subjects [23,24,25].

Hand odor collection would allow for a more rapid screening of individuals for disease states versus breath analysis methodologies, which require large volumes of exhaled breath to be collected for disease state determination. The analysis of hand odor also proves less invasive than the nasopharyngeal swabbing incurred for various RT-qPCR and antigen diagnostic tests. Additionally, hand odor swabs can be treated as a “clean sample” requiring no sample cleanup or liquid extraction prior to analysis by analytical instrumentation.

For these reasons, this research seeks to leverage the VOC detection of human hand odor using headspace solid phase microextraction (HS-SPME) sampling and gas chromatography-mass spectrometry (GC-MS) analysis to identify chemical biomarkers that may distinguish between SARS-CoV-2-positive and -negative individuals. The use of chemometric tools allows for efficient data manipulation and machine learning to develop sPLS-DA predictive models. This is the first in a set of papers that lays the groundwork for a non-invasive diagnostic approach for SARS-CoV-2. It is the belief of the researchers that the further expansion and successful implementation of this approach will also result in a framework for the development of rapid diagnostic tools for other diseases.

## 2. Materials and Methods

### 2.1. Human Hand Odor Collection

#### 2.1.1. Sample Collection Details

Human hand odor samples were collected at three different time periods corresponding to the height of the Delta variant and the two Omicron SARS-CoV-2 subvariants (Table 1). The samples collected during late 2021 (Delta variant’s dominance) were repurposed from a separate task. As such, the sample sets collected during 2021 and 2022 were collected using different collection procedures.

All samples were collected by researchers at the Penn Acute Research Collaboration- Penn Presbyterian Medical Center (PARC-PPMC) in Philadelphia, PA. The samples were collected from a patient population present in the Emergency Department at the time of sampling. All samples were collected under the approval of the University of Pennsylvania’s Institutional Review Board (IRB# 848819). The VOC analysis of the collected samples was conducted by researchers at Florida International University in Miami, Florida.

#### 2.1.2. Late 2021 Hand Odor Samples

Prior to the sample collection, the patients were asked to “cleanse” their hands by wiping them with a WaterWipes^®^ wipe. They were then instructed to dry their hands with a paper towel or air-dry them. Samples were collected using sterile medical gauze. The gauze packaging was opened by the researcher without touching the gauze. The patient was instructed to (1) grab the exposed gauze and ball it up in their hands, (2) hold it in between their closed palms for approximately 10 s, and then (3) wipe both hands with the gauze before (4) placing it into a plastic specimen bag.

The specimen bag was labeled with a patient identifier. The hand odor sample was collected in conjunction with an underarm odor sample and a method control (blank) sample. All samples were packaged into separate specimen bags. The hand odor and control samples were used in this study; the underarm odor samples were not. The samples were stored below 3 °C until they were transferred to Florida International University for SPME odor extraction and subsequent GC-MS analysis.

#### 2.1.3. Early-2022 and Mid-2022 Hand Odor Samples

Samples were collected using pre-treated sterile gauze [26]. The researcher prepared for the collection procedure by opening a vial containing the gauze and removing it with a pair of clean tweezers. The gauze was placed into the patient’s hand, and the patient was instructed to (1) wipe both palms fully, (2) hold the gauze between their closed palms for 30 s, and then (3) ball-up the gauze and push it through the opening of the vial (held by the researcher). The researcher then resealed the vial and placed it into an aluminum barrier bag. This bag and the vial were labeled with patient information and then stored at −20 °C until they were ready for analysis. The samples were transferred into FIU’s custody for chemical analysis and were delivered along with three method blank samples per shipment. The samples were generally received monthly during active sample collection periods.

### 2.2. Confirmation of SARS-CoV-2 Infection

In this study, 95 individuals were tested and diagnosed for SARS-CoV-2 infection using the Roche cobas^®^ SARS-CoV-2 Duo Test for use on the cobas^®^ 6800/8800 systems (Roche Diagnostics, Basel, Switzerland). The healthcare provider collected nasopharyngeal swab samples from all 95 patients for this PCR diagnostic test, which is the standard SARS-CoV-2 diagnostic method used by PPMC. Positive results were reported, as per the FDA EUA instructions for use, and signify the presence of SARS-CoV-2 RNA [27]. Three (3) individuals were tested using the Abbott BinaxNOW antigen test (Abbott Laboratories, Chicago, IL, USA), which consisted of a self-collected anterior nasal swab specimen. The results of this test indicated whether the nucleocapsid protein antigen of SARS-CoV-2 was identified [28]. Finally, the remaining four (4) individuals were not tested at PPMC; while the results of their SARS-CoV-2 test were relayed to the researchers, the sample specimen and test type remain unknown.

### 2.3. Patient Demographics

All patients were asked to verbally disclose the following information to the researchers: age, race/ethnicity, sex at birth, symptomology, vaccination status, and chief health complaint (what brought them to the hospital). This information can be found in the Appendix A in Table A1.

### 2.4. Preparation of Collection Materials

#### 2.4.1. Vial Cleaning Procedure

The utilized 10 mL vials were (1) sonicated in a mild, soapy cleaning solution, followed by scrubbing and rinsing under tap water, (2) sonicated in a water bath, (3) sonicated in a bath of deionized water, (4) rinsed with acetone, and (5) baked at 105 °C for a minimum of an hour prior to being used to store collection materials. The sonicator was set to a 30 min cycle and held at 30 °C at each instance of use.

#### 2.4.2. Pre-Treatment of Cotton Gauze

Sterile cotton gauze pads (100% cotton) served as the sorbent medium for the collection of Early-2022 and Mid-2022 human hand odor samples. Gauze pads of 2″ by 2″ and eight-ply density (Dukal Corporation, Syosset, Oyster Bay, NY, USA) were acquired and treated to a cleaning procedure prior to their use in sampling. The undergone pre-treatment procedure entails spiking 1 mL of HPLC-grade methanol (Fisher Chemical, Bridgewater, NJ, USA) onto the sterile gauze and baking the substrate for a minimum of one hour at 105 °C.

#### 2.4.3. Storage & Containment

Pre-treated cotton gauze squares were stored in the cleaned 10 mL vials. The vials were sealed and secured with parafilm around the screw cap opening. All vials were labeled to collect information regarding: (a) Sample #, (b) Date, (c) Sex at Birth, and (d) SARS-CoV-2: Positive or Negative.

### 2.5. HS-SPME-GC-MS Method

The samples were placed in a digital heating bath set at 50 °C and left to equilibrate for 24 h. After this period, a clean 50/30 µm divinylbenzene/ carboxen/ polydimethylsiloxane (DVB/CAR/PDMS) SPME fiber was exposed to the headspace of the hand odor samples at a 1-inch fiber exposure setting. After 15 h, the SPME fibers were unexposed and removed from the sample headspace.

Analytes on the SPME fibers were thermally desorbed at 270 °C for 5 min (2-inch fiber height) into the heated inlet of the GC (Agilent 8890; Agilent Technologies, Santa Clara, CA, USA). A splitless injection method with a 1 mL/min column flow was implemented on an HP5-MS UI capillary column (15 m × 0.250 mm × 0.25 µm I.D.; Agilent Technologies). UHP Helium was used as the carrier gas. The oven temperature parameters started at 40 °C (1.25 min hold), increased to 165 °C (5 °C/min rate), and concluded at 270 °C (30 °C/min rate). The total method runtime was 29.75 min. A mass spectrometer (MS) (Agilent 5977B MSD; Agilent Technologies) with an electron impact ionization (EI) source and quadrupole mass analyzer was used, with the following parameters: the MS source was maintained at 230 °C, the MS Quad at 150 °C, the transfer line at 280 °C and the EI source at 70 eV. Samples collected between June 2021 and May 2022 were analyzed using a scan range of *m*/*z* 50–550. Samples analyzed between July 2022 and October 2022 were analyzed using a 45–400 *m*/*z* scan range.

### 2.6. Data Pre-Processing

Collected human hand odor samples were analyzed using the described HS-SPME-GC-MS method (Section 2.5). The resulting datafiles were retention time-aligned, and peak matching was performed across the dataset. Following this procedure, the files were background-subtracted using the associated control sample. The background-subtracted samples were separated into four subgroups: (1) Late-2021 SARS-CoV-2-positive samples, (2) Early-2022 SARS-CoV-2-positive samples, (3) Mid-2022 SARS-CoV-2-positive samples, and (4) SARS-CoV-2-negative samples.

Within the SARS-CoV-2-positive subgroupings, the variance of each peak of interest was determined. Peaks of interest that demonstrated zero variance were removed from consideration from the sample set as a whole. Additionally, peaks corresponding to background interferents present in the method blank samples or identified as a recurring non-target such as methylene chloride or siloxanes peaks (present in the column phase and SPME fiber used) were removed from the sample set. The peaks of interest were able to be filtered down to include compounds eluting prior to the 21 min mark. The total ion chromatogram (TIC) peak areas were log10-transformed prior to modeling to reduce the skewedness in the data; all values of 0 were set to 1 before applying the transformation.

### 2.7. Statistical Analysis

Sparse partial least squares discriminant analysis was used to form predictive models indicative of a patient’s SARS-CoV-2 infection status. There were four models developed using the log10-transformed TIC peak areas of the 40 features of interest. The first model demonstrates the outcome of informing a model using multiple sample collection timeframes (2021–2022), and models 2–4 demonstrate predictive models informed by a singular SARS-CoV-2 sample collection timeframe, which is believed to relate to a single variant’s dominance in the population.

sPLS-DA modeling was performed using the “mixOmics” packages in R (Version 3.6.1, Vienna, Austria) [29]. In all cases, the sPLS-DA models were informed by an equal number of positive and negative samples. The lesser of the two was used as the defined class size, and the larger group was randomly sampled to provide an equal number of samples from each class. The models were cross-validated using a five-fold cross-validation, repeated 200 times. This resulted in a random division of the samples into 80% training set and 20% test set.

## 3. Results

### 3.1. sPLS-DA Modeling for All Timeframes

Following the pre-processing procedure detailed in Section 2.6, the sample set contained 40 features of interest. This sample set was log10-transformed and used to conduct modeling of the SARS-CoV-2-positive and -negative hand odor samples. The sPLS-DA model of the hand odor samples demonstrates the class grouping of the positive and negative samples (Figure 1). The loading contributions of each feature of interest can be found in Figure A1 and Figure A2.

The sPLS-DA model shown in Figure 1 depicts the individual human hand odor samples collected from SARS-CoV-2-positive (*n* = 56) and SARS-CoV-2-negative (*n* = 46) individuals. These samples were collected across a timespan that included the dominant presence of both the Delta and Omicron variants of the SARS-CoV-2 virus. The model depicted (Figure 1) was cross-validated using a five-fold cross-validation, which was repeated 200 times; this procedure randomly sampled the positive samples to construct models informed by *n* = 46 positive and negative samples. Following this cross-validation, the model was seen to yield an accuracy of 75.8% (±0.4); the model correctly predicted the SARS-CoV-2 infection status of a sample in 75.8% (±0.4) of attempts (95% CI). This performance breaks down to a sensitivity/true positive rate (TPR) = 81.8% (±0.5), specificity/true negative rate (TNR) = 69.7% (±0.6), false positive rate (FPR) = 30.3% (±0.6), false negative rate (FNR) = 18.2% (±0.5), positive predictive value (PPV) = 73.0% (±0.4), and negative predictive value (NPV) = 79.4% (±0.5).

### 3.2. sPLS-DA Modeling for Individual Timeframes

Figure 2 demonstrates the modeling of Late-2021 positive samples (*n* = 20) and negative SARS-CoV-2 infection samples (*n* = 46). The model depicted (Figure 2) was cross-validated using a five-fold cross-validation, which was repeated 200 times; this procedure randomly sampled the negative samples to construct models informed by *n* = 20 positive and negative samples. Following this cross-validation, the model yielded an 86.7% (±0.6) accuracy rate (95% CI). This performance breaks down to a sensitivity/TPR = 84.2% (±0.6), specificity/TNR = 89.2% (±0.9), FPR = 10.8% (±0.9), FNR = 15.8% (±0.6), PPV = 89.0% (±0.9), and NPV = 85.0% (±0.5).

Figure 3 demonstrates the modeling of Early-2022 positive (*n* = 13) and negative SARS-CoV-2 infection samples (*n* = 46). The model depicted (Figure 3) was cross-validated using a five-fold cross-validation, which was repeated 200 times. The negative samples were randomly sampled to construct models informed by *n* = 13 positive and negative samples. The cross-validated model yielded a 64.4% (±1.0) accuracy rate. This performance breaks down to a sensitivity/TPR = 73.7% (±0.7), specificity/TNR = 55.0% (±1.7), FPR = 45.0% (±1.7), FNR = 26.3% (±0.7), PPV = 62.7% (±1.0), and NPV = 67.1% (±1.0).

Figure 4 demonstrates the modeling of Mid-2022 positive (*n* = 23) and negative SARS-CoV-2 infection samples (*n* = 46). The model depicted (Figure 4) was cross-validated using a five-fold cross-validation, which was repeated 200 times; this procedure randomly sampled the negative samples to construct models informed by *n* = 23 positive and negative samples. Following this cross-validation, the model yielded an 83.6% (±0.8) accuracy rate. This performance breaks down to a sensitivity/TPR = 90.5% (±0.9), specificity/TNR = 76.7% (±1.0), FPR = 23.3% (±1.0), FNR = 9.5% (±0.9), PPV = 79.7% (±0.8), and NPV = 89.3% (±1.0).

### 3.3. Identification of Features of Interest

Out of the 40 features of interest used to inform the sPLS-DA models, the corresponding compound identifications of 14 were determined. These compounds (listed in Table 2) were identified through manual confirmation using externally run reference standards.

## 4. Discussion

### 4.1. SARS-CoV-2 Infection Diagnostic Model Performance

SARS-CoV-2-infected human odor expression was investigated through the collection and analysis of hand odor samples. Hand odor samples were collected from 102 individuals; of these participants, 56 were SARS-CoV-2-positive and 46 were negative for SARS-CoV-2 at the time of donation. PCR tests were used as the primary confirmatory method in diagnosing SARS-CoV-2 infection status. The samples were collected using sterile gauze and analyzed using HS-SPME-GC-MS. The resulting human hand odor profile was analyzed using the previously described data pre-processing methods, resulting in a key list of 40 features of interest. Class separation (SARS-CoV-2-positive vs. -negative) was modeled using the log10-transformed total ion chromatogram peak areas of the features of interest. The resulting sPLS-DA modeling of all of the collected SARS-CoV-2-positive and -negative samples (Figure 1) demonstrated class grouping within the negative and positive samples. Although a defined separation of classes was not observed, the model reflected a moderate performance ability with 75.8% accuracy when compared to other screening and diagnostic techniques such as antigen tests and detection canines (as summarized in a previous publication [12]). This rate reflected a sensitivity (TPR) of 81.8% and a specificity (TNR) of 69.7% for predicting the SARS-CoV-2 status. The performance of each model based on a repeated five-fold cross-validation (*n* = 200) can be seen in Table 3 below.

The SARS-CoV-2-positive samples were further separated into Late-2021, Early-2022, and Mid-2022 samples and are suspected to correspond to Delta, Omicron BA.2, and Omicron BA.5 variants, respectively. The variant type was not confirmed; it was assigned based upon the variant that was dominant in the collection area (Philadelphia, Pennsylvania) at the time of sample donation. This assignment was conducted using the Centers for Disease Control and Prevention’s variant proportional COVID data reporting platform [40]. The Late-2021 SARS-CoV-2-positive samples were modeled against the SARS-CoV-2-negative samples, revealing an increased separability between the classes (Figure 2) compared to that demonstrated in Figure 1. This observed separation translated into an improved performance in correctly identifying SARS-CoV-2-infected sample status (86.7% accuracy). Similarly, the modeling of Mid-2022 SARS-CoV-2 (Figure 4)-positive samples reflected an increase in the rate of accuracy to 84.4%. However, the individual modeling of Early-2022 SARS-CoV-2-positive samples vs. SARS-CoV-2-negative samples (Figure 3) resulted in a reduction in discriminatory power, with a reduced rate of accuracy at 64.4%.

The individual modeling of samples by variant indicates that sPLS-DA models trained on samples collected within the dominance of the Delta or Omicron BA.5 variants yielded a greater predictive power than models trained using samples collected during the height of Omicron BA.2 or informed by multiple variant timeframes. The change in model performance in relation to SARS-CoV-2 variant dominance is an interesting phenomenon which suggests a difference in the underlying VOC expressions between SARS-CoV-2 variants. This variant-dependent expression of SARS-CoV-2 via volatile emanations is a topic of future study and will be further investigated by the researchers. Despite the suggestion of variant dependency in SARS-CoV-2 expression, the predictive power of the sPLS-DA model informed by all samples collected between June 2021 and October 2022 suggests that there is an underpinning commonality in the human odor expression associated with SARS-CoV-2 infection.

This work relied upon the sPLS-DA models which were informed by 40 features of interest. The researchers suggest that these compounds be considered potential biomarkers for the expression of SARS-CoV-2 via human odor. Of the 40 features of interest utilized in the machine learning models, 14 compounds have been identified (Table 2). Many of these compounds consist of aldehydes, alcohols, ketones, and other functional groups that are common in human scent [26]. The authors were able to identify instances of the published presentation of these compounds in human scent samples for 12/14 compounds (Table 2). The previous reporting of two compounds (diacetone alcohol and 2-phenoxyethanol) in skin emanations was not determined. However, it is noted that 2-phenoxyethanol has been detected in human feces and saliva samples, while diacetone alcohol has been identified as a secondary metabolite and is believed to play a role in signaling processes; both have been identified in human samples, but to the authors’ knowledge, they have not been reported in human scent samples.

### 4.2. Limitations and Future Perspective

Due to the wide performance range of the antigen tests currently on the market, there remains a fair possibility that the three individuals tested by an antigen test alone were not accurately accounted for (positive or negative). While the PCR test is the current gold standard for SARS-CoV-2 detection and diagnostics, it is an imperfect tool as well. As with most diagnostic tools, it is important to acknowledge the possibility of false positive and false negative results produced by PCR tests. Additionally, the tests are bound to their own detection limitations. Participants who tested as “negative” using a PCR test may have attained this result due to a low viral load or stage of infection as opposed to a “truly negative” SARS-CoV-2 infection status.

In addition to concerns over ground truth SARS-CoV-2-infected patient status, there are concerns over previous contractions of SARS-CoV-2 and the possibility that patients were afflicted by long-COVID at the time of sample collection. Recent work in the field of biomedical scent detection has suggested that canines trained to detect SARS-CoV-2-infected positive samples may also be able to detect long-COVID, which is the persistent effects of COVID-19 that remain beyond PCR test positivity (post-COVID affliction) [41]. These new findings suggest that there is more similarity between long-COVID odor presentation and SARS-CoV-2-positive odor presentation than there is between long-COVID and SARS-CoV-2-negative presentations. Due to the difficulty in properly determining previous SARS-CoV-2 infection (individuals may have been asymptomatic or have gone untested during a period of illness), the researchers did not actively seek to identify and remove individuals who may have been suffering from long-COVID. This consideration is viewed as a reasonable approximation of the real-world application of this work, as patients would be subject to SARS-CoV-2 screening regardless of long-COVID status.

The authors intend to continue their investigation of the VOC expression of SARS-CoV-2 infection as it relates to human hand odor. This work served as an initial investigation of the probative value of human hand odor as a medium for SARS-CoV-2 infection status determination. The authors intend to continue this discussion in a secondary manuscript wherein the influence of the suspected SARS-CoV-2 variant is evaluated as a factor driving class separation. Additionally, the separability of SARS-CoV-2 variants will be examined.

The presented methodology of using human hand odor as a potential diagnostic medium is a promising prospect for an adaptive field-deployable technology. While the authors note that the infrastructure does not currently exist for the wide application of this technology, they believe that human hand odor analysis is a viable diagnostic medium that has the capacity to be expanded upon and adapted into a fieldable methodology. The greatest benefit of this technology would reside in mass-screening venues such as airports or other readily visited areas with elevated concerns of transmission. This begins with the adoption of portable GC-MS instrumentation, allowing for lower-cost installations. This methodology could also open a pathway for the continual improvement of the technology through the periodic introduction of new SARS-CoV-2-positive and -negative samples (verified by PCR testing). As status is confirmed, more samples can be added to the model, allowing for continual training; as SARS-CoV-2 variants evolve, so will the model. The adoption of a machine learning-backed approach provides a route to teaching and improving diagnostic models at a rate that is competitive regarding the continual evolution of our targeted virus.

## 5. Conclusions

This work demonstrated that HS-SPME-GC-MS could be used jointly with a developed sPLS-DA model to predict the SARS-CoV-2 infection status of a patient. Furthermore, the developed method shows that using odor samples from the hands of subjects is sufficient to serve as a potential diagnostic medium. The developed model displayed a 75.8% accuracy. The performance rate was seen to be similar for the sensitivity rate (81.8%) and specificity rate (69.7% correct) for correctly predicting SARS-CoV-2 infection status, implying that this technology has the capacity to be competitive with antigen testing in terms of performance metrics. This model displays an improved overall performance ability when compared to the model informed by a timespan-specific subset corresponding to the dominant Omicron BA.2 and reflects decreased performance when compared to the models trained on the Delta and Omicron BA.5 correlated sample subsets. The computational model was informed by 40 features of interest; at this time, the compound identities of 14 out of 40 features have been identified and confirmed using certified reference standards. This noninvasive, low-preparatory-analysis scheme has the potential to be continually informed to evolve alongside the SARS-CoV-2 virus. The further optimization of this diagnostic analysis could facilitate the rapid screening and mitigation of disease spread and could potentially be adapted to other infectious diseases with expressed human odor profile variations.

## Figures and Tables

**Figure 1 diagnostics-13-00707-f001:**
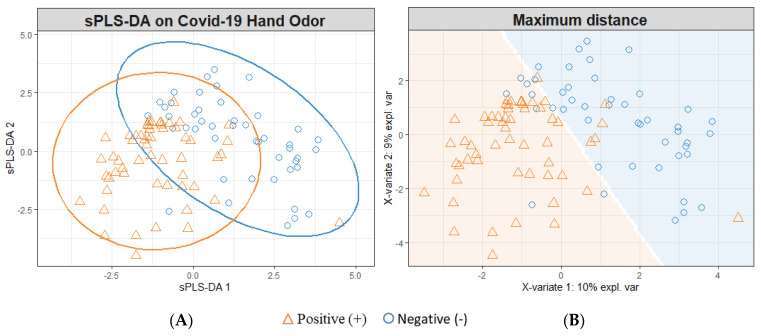
All-Timeframes SARS-CoV-2 sPLS-DA Model; (+) SARS-CoV-2 Samples vs. (−) SARS-CoV-2 Samples. (**A**) Individual sample plot; ellipses = 95% CI. (**B**) Individual sample plot; background delineates class assignment.

**Figure 2 diagnostics-13-00707-f002:**
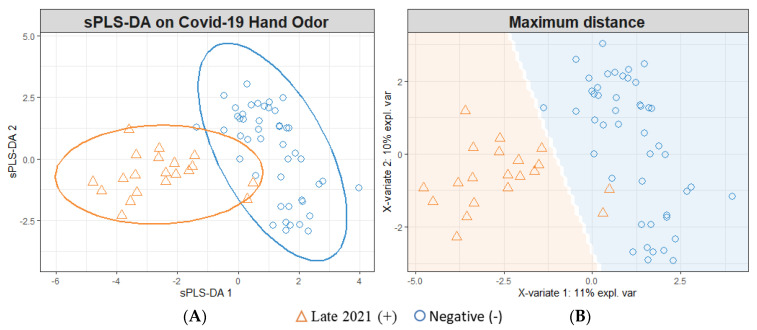
Late-2021 sPLS-DA Model; (+) Late-2021 Hand Odor Samples vs. (−) SARS-CoV-2 Samples. (**A**) Individual sample plot; ellipses = 95% CI. (**B**) Individual sample plot; background delineates class assignment.

**Figure 3 diagnostics-13-00707-f003:**
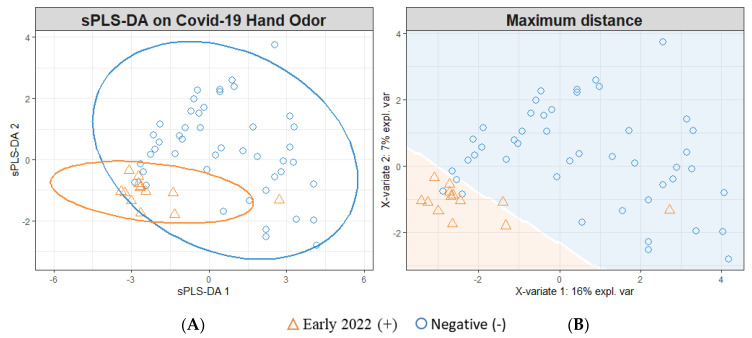
Early-2022 sPLS-DA Model; (+) Early-2022 Hand Odor Samples vs. (−) SARS-CoV-2 Samples. (**A**) Individual sample plot; ellipses = 95% CI. (**B**) Individual sample plot; background delineates class assignment.

**Figure 4 diagnostics-13-00707-f004:**
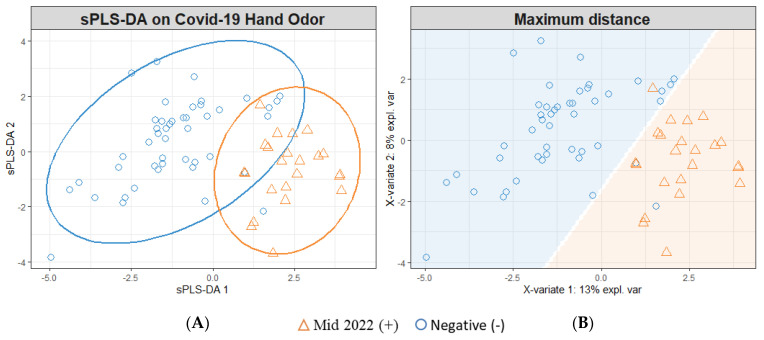
Mid-2022 sPLS-DA Model; (+) Mid-2022 Hand Odor Samples vs. (−) SARS-CoV-2 Samples. (**A**) Individual sample plot; ellipses = 95% CI. (B) Individual sample plot; background delineates class assignment.

**Table 1 diagnostics-13-00707-t001:** SARS-CoV-2-Positive and -Negative Samples.

SARS-CoV-2 Infection Status	Timeframe	Collection Timespan	Dominant SARS-CoV-2 Variant at Time of Collection	Number of Samples
Positive	Late 2021	June 2021–October 2021	Delta	20
	Early 2022	February 2022–May 2022	Omicron BA.2	13
	Mid 2022	July 2022–September 2022	Omicron BA.5	23
Negative	2021–2022	September 2021–October 2022	Delta/Omicron BA.2/ Omicron BA.5	46

**Table 2 diagnostics-13-00707-t002:** Compounds contributing to the discrimination of SARS-CoV-2 infection.

Retention Time (15 M HP5-MS)	CAS#	Compound of Interest	Reported Presence in Human Skin Emanations
3.109	00123-42-2	Diacetone alcohol	-
4.033	00100-42-5	Styrene	[16,30]
6.577	03777-69-3	2-Pentylfuran	[16,31]
7.897	00122-78-1	Phenylacetaldehyde	[32]
9.552	01120-21-4	Undecane	[16,24,25,26,31,33,34]
10.366	00111-11-5	Methyl caprylate	[24,25,26,31,34,35]
11.235	18829-56-6	trans-2-Nonenal	[16,24,25,31,35]
11.679	00143-08-7	1-Nonanol	[24,31,34,35]
12.561	00112-31-2	Decanal	[16,24,25,26,30,31,33,35]
12.836	00122-99-6	2-Phenoxyethanol	-
17.798	00112-54-9	Dodecanal	[24,26,30,31,35]
18.841	00689-67-8	6,10-Dimethyl-5,9-undecadien-2-one-(E)	[16,24,25,26,30,31,33,34,35]
19.366	00112-53-8	1-Dodecanol	[36,37]
21.668	00143-07-7	Dodecanoic acid	[24,30,31,35,38,39]

**Table 3 diagnostics-13-00707-t003:** Overall and Variant-Specific sPLS-DA Performance.

sPLS-DA Model Correct Prediction Rates *				
Time Range	2021–2022	Late 2021	Early 2022	Mid 2022
Suspected Variant	All	Delta	Omicron BA.2	Omicron BA.5
Accuracy	75.8% (±0.4)	86.7% (±0.6)	64.4% (±1.0)	84.4% (±0.8)
Sensitivity (TPR)	81.8% (±0.5)	84.2% (±0.6)	73.7% (±0.7)	90.5% (±0.9)
Specificity (TNR)	69.7% (±0.6)	89.2% (±0.9)	55.0% (±1.7)	76.7% (±1.0)
FNR	18.2% (±0.5)	15.8% (±0.6)	26.3% (±0.7)	9.5% (±0.9)
FPR	30.3% (±0.6)	10.8% (±0.9)	45.0% (±1.7)	23.3% (±1.0)
PPV	73.0% (±0.4)	89.0% (±0.9)	62.7% (±1.0)	79.7% (±0.8)
NPV	79.4% (±0.5)	85.0% (±0.5)	67.1% (±1.0)	89.3% (±1.0)

* Error Rate = 100% (Correct Prediction Rate).

## Data Availability

The dataset associated with this study has been submitted to the National Institute of Health’s Scientific Data Sharing platform. Access to the dataset can be requested through https://sharing.nih.gov/, accessed on 16 January 2023.

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
