# Peer review of "Investigating the Use of SARS-CoV-2 (COVID-19) Odor Expression as a Non-Invasive Diagnostic Tool—Pilot Study"

_diagnostics, 2023, doi:10.3390/diagnostics13040707_

Round 1

Reviewer 1 Report

The contribution of the current manuscript is not sufficient for publication in this Journal. Thus,  I regret to inform you that I have decided against publishing your manuscript.

Author Response

We regret to find that you do not support the publication of this manuscript in MDPI Diagnostics; nevertheless, we would like to thank you for your time and effort in reading and refereeing the contents of our work. 

Reviewer 2 Report

Due to the insufficient number of samples analyzed in individual groups of positive patients (less than 30 samples taken), which does not provide statistical power for the conclusions drawn, the title of the work should be: "Investigating The Use Of COVID-19 Odor Expression As A 2 Non -Invasive Diagnostic Tool – Preliminary (Pilot) Study”

Line 42 - if the sensitivity for RT-qPCR is given, it is appropriate to also give the range for antigen tests.

Line 98 - sex at birth or biological sex ?

Line 99 - chief health complaint or main disease symptom ?

Line 185 - could explain why peak areas were log10 transformed prior to model ling.

Line 226 - please treat the model as a diagnostic method and provide diagnostically significant parameters: sensitivity, specificity of the model with the false positive and false negative rate as well as PPV and NPV. The same for individual study groups and variants in the tables. Extracting these data will increase the comprehensibility and assessment of the diagnostic usefulness of the method.

Materials and Methods - there is no description, the source of the swab, the method of isolation of the genetic material of the virus, the name of the test used and the key parameters used to identify patients positive by the RT-qPCR method. It is also necessary to provide information whether all groups were tested in the same way and with the same test, not the technique. Please add me please.

Please provide information whether the diagnostic model will continue to be fed with further well-characterized samples in order to increase the value of the desired diagnostic parameters.

Author Response

First of all, we would like to thank you for taking the time to read through and provide thoughtful notes on our manuscript. Enclosed are the responses to your comments on the current manuscript. 

Reviewer 3 Report

The topic is very important and I also appreciate the quality of this manuscript.

The research question is clear, the method is suitable, and the data analysis method is very convincing.

The research results are good presented and reliable.

However, I suggest the authors edit some small content as follows:

1. The Introduction should present the advantages and disadvantages of the diagnostic methods that have been and are being used, as well as the need for a rapid diagnostic method with high reliability.

2. The author needs to distinguish between SARS-CoV-2 infection and COVID-19 disease. Because SARS-CoV-2 infected people may have no symptoms. Authors should avoid saying: "COVID-19 infected individuals" but should replace with "SARS-CoV-2 infected individuals"

3. Therefore, it is necessary to add to Results a part about the sensitivity and specificity of this diagnostic method for symptomatic and asymptomatic people.

4. The authors should add the limitations of the research method, especially the reproducibility for the following studies and the practicality of application in the limited conditions of diagnostic facilities.

Author Response

(The authors gave the same response as above.)

Round 2

Reviewer 1 Report

This is a well-written, well-organized and well-illustrated paper. It presents the results of original research and makes a valuable contribution to knowledge.

Reviewer 3 Report

Thank you very much for your responses 

I have no further comments

Congrats!